# Extracting user profile via large language models and ontologies

**Pegah Safari**[ID]**, Mehrnoush Shamsfard**[ID]*

Faculty of Computer Science and Engineering, Shahid Beheshti University, Tehran, Iran

* m-shams@sbu.ac.ir

## Abstract

Extracting user-specific profiles that include general personal information, such as hobbies, occupation, or age, is a valuable asset for systems like recommendation engines or personalized chatbots. Currently, most approaches moved toward exploiting the capabilities of large language models (LLMs) on rich languages while less-resourced languages still present significant opportunities for exploration and improvement. In this research, we present a multi-step approach for profile extraction from dialogue systems in Persian as a use case. Through an extensive set of experiments and analyses on various models, we show that LLMs struggle with these languages due to limited language-specific resources and complex linguistic structures. To address this, we propose a hybrid method that combines techniques such as slot filling, in-context learning, and ontology-based inference. Our final results demonstrate a significant improvement over current state-of-the-art models, including LLMs with few-shot examples and even their fine-tuned version. Our method achieves an F-score of 90.46, outperforming GPT-4o and Llama-3-70B by an absolute difference of 17.08 and 24.51 respectively. Our system can also detect inconsistencies in presented information in which our performance substantially exceeds the best performance of GPT-4o and Llama-3-70B with an accuracy of 92%. It is 21% absolutely better than GPT-4o and 47% better than Llama.

## 1 Introduction

Developing an effective Natural Language Understanding (NLU) component to serve as the backbone of dialogue systems has long been a topic of significant interest [1–3]. This component heavily relies on natural language processing techniques to extract *structured* information from *unstructured* user input [4–6]. In this research, we focus on a specific type of NLU task: extracting user profile information from chit-chat dialogues [7]. User profile information includes, but is not limited to, attributes such as gender, age range, location, and marital status. This information is valuable in a variety of applications, including personalized dialogue systems [8–10], automatic persona detection for persona-grounded chatbots [11], online therapy conversations

**Data availability statement:** Data in this research is available in Zenodo repository with following DOI: 10.5281/zenodo.15839326 also the codes are available in GitHub: https://github.com/phsfr/Multistep-Profile-Extraction-LLM-Fa.

**Funding:** The author(s) received no specific funding for this work.

**Competing interests:** No authors have competing interests.

[12,13], and recommendation systems [14–16]. For example, in the context of therapy sessions, understanding the demographic background of a client enables the system to better identify relevant issues and match the client with the most appropriate expert [17,18].

Focusing on natural language understanding in conversational systems, there has been considerable progress in using large language models (LLMs) for such tasks [19–21]. Despite their impressive results in high resource languages, it is not clear whether a standalone LLM can be as effective in languages that suffer from data scarcity. In this work, we examine Persian as a representative case of such languages. Persian is primarily spoken in Iran, as well as parts of Afghanistan and Tajikistan, with an estimated 110 million speakers [22]. Notably, the past decade has witnessed a significant increase in Persian content on online platforms. For instance, the number of sellers on Digikala (a leading e-commerce platform in Iran) grew by over 23.6% in 2023 compared to the previous year [23].

Despite recent efforts to curate datasets for Persian NLU [24–29], we still suffer from lack of resources, and it is not surprising to see that even large language models do not perform as well in Persian compared to high-resource languages. For example, unlike recent findings in English [30], we observe that standalone LLMs do not perform as well as encoder-only language models for intent detection and slot filling. We show that our proposed model surpasses the F-score of GPT-4o and Llama by a substantial margin. Also, we demonstrate that GPT-4o fails to accurately extract certain key user profile attributes from Persian utterances. To address these challenges, we introduce a multi-step approach that combines classical slot-filling techniques [31] with few-shot prompting to effectively extract user profile information. Fig 1 illustrates the overall architecture of our model pipeline.

In the first step, we employ a slot-filling method to extract phrases that may indicate user profile attributes. In parallel, we prompt a large language model to infer additional profile-related information from the same utterance. In the second step, we feed the outputs from both the slot filler and the LLM into a Persian ontology that incorporates socio-cultural and contextual life knowledge to infer the final user profile information. Importantly, our ontology is also capable of detecting inconsistencies in user-provided data, such as identifying a mismatch between a user claiming to live in the city of Tehran and with Nigeria as the country of their residence. Our method achieves an accuracy of 92%, outperforming *GPT-4o* by an absolute margin of 21%, and Llama-3 by 47%.

The main contributions of this work are summarized as follows:

- We propose a multi-step user profile extraction method as a crucial component of a NLU system that leverages both classical methods such as slot filling and ontology, as well as more advance methods such as few-shot prompting with large language models.

- Our final results demonstrate a significant improvement over two prominent large language models. Our user profile extraction achieves a f-score of 90.46 which is at least 20.1 scores higher than the best performing LLM. We also achieve a similar gain in finding out inconsistencies in extracting user profile information.

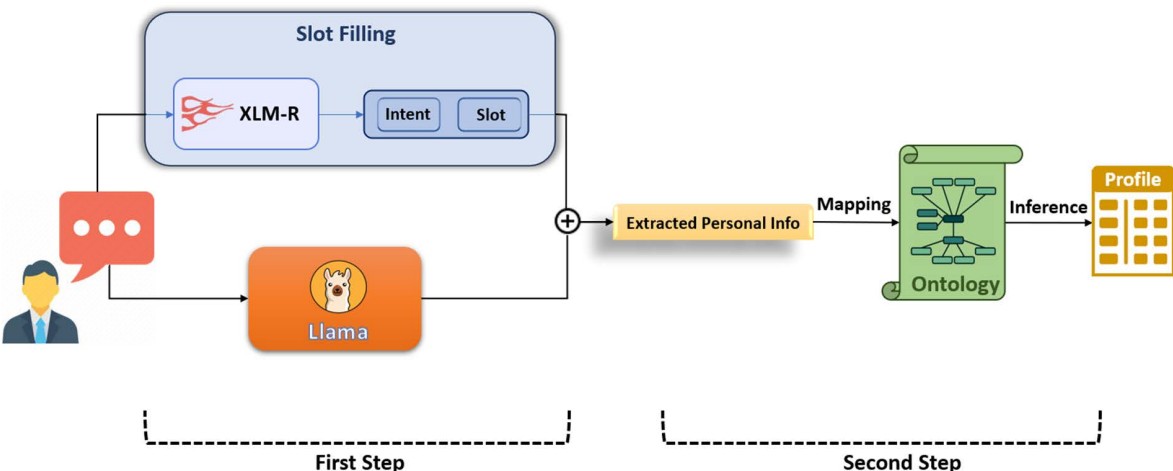

**Fig 1. Overall architecture of our proposed multi-step approach for user profile extraction.**

- We conduct comprehensive experiments and analytical studies on the zero-shot and fine-tuning results of recent LLMs, including the GPT family [32], Llama [33], and Phi [34]. Our findings reveal biases and shortcomings in LLMs, as well as a tendency to take any applicable value as slot values, leading to inflation of recall at the cost of precision.

The rest of the paper is organized as following: §2 describes our approach, §3 presents the experimental setting, and §4 shows the final results. In §5 and §6, we provide detailed analysis and discussion of our results versus state-of-the-art models. Finally, after reviewing related work in §7, we conclude in §8.

## 2 Materials and methods

Focusing on extraction of user profile information in chit-chat dialogues, we can formulate this task into extracting a diverse set of of personal information such as age, gender, marital status, number of children/siblings, residential city/ province, hobby, and occupation [7]. So first, in this section, we describe our proposed methodology, followed by a discussion of the ethical considerations involved.

### 2.1 Methodology

Given an input sentence with $n$ tokens ($X = x_1, \cdots, x_n$ ), we aim to extract profile entries that can also be associated with the topic of the utterance. As shown in Fig 1, we employ a multi-step process, where a slot-filling model based on sequential tagging with the BIO schema [31] extracts phrases or words that correspond to user specifications (e.g., name or city). At the same time, using a LLM with few shot prompting (aka in-context learning), we extract more nuanced information that may not be as explicitly stated. The next step involves mapping all extracted information and LLM judgments to an ontology. Based on the inferences made from the ontology, we populate the user's final profile. Additionally, the ontology can identify potential semantic conflicts between the presented and inferred information. We explain each of these steps in detail below.

**2.1.1 Step 1.1: Slot filling.** Given an input sentence $X$ comprising $n$ tokens: $X = x_1, \cdots, x_n$ , our slot filling model assigns labels $Y = y_1, \cdots, y_n$ to each token where $y_i$ belongs to either an O (not a label) tag, at the beginning of a slot type (such as B-Gender) or inside a slot type (such as I-Gender). We use an encoder-only language model such as XLM-Roberta [35] to convert the input sentence $X$ to vector representations $H = h_1, \cdots, h_n \in \mathbb{R}^d$ and use a softmax layer with weight matrix $W \in \mathbb{R}^{c \times d}$ to calculate probabilities of each of the $c$ slot types. We run inference with a greedy classifier and use a cross-entropy loss function to find model parameters during training.

**2.1.2 Multitask learning with intent detection.** We leverage multi-task learning by incorporating a general intent detection task during training to improve the accuracy of our model. The mutual information between detected intents and their associated slot types benefits the slot-filling when trained jointly [31,36].

Considering $k$ intent types, we use the same encoder-only layer used for slot filling and extract a pooled representation $\bar{H} = \text{pool}(h_1, \cdots, h_n)$. We then apply an intent classifier: $z = W \cdot \text{Dropout}(\bar{H}) + b$ where $Dropout(.)$ represents the dropout function applied during training, $W$ is the weight matrix ($W \in \mathbb{R}^{k \times d}$), $b$ is the bias vector ($b \in \mathbb{R}^k$) and $z$ is intent logits. At the top layer, we perform two separate classification tasks to evaluate model performance in both single-intent and multi-intent modes. For single-intent setting, we apply a softmax function over the output logits and use the cross-entropy loss during training. In the multi-intent mode, we use a sigmoid activation for independent label predictions and train the model with a binary cross-entropy loss.

**2.1.3 Step 1.2: Few-shot prompting.** The diversity of information types presented in dialogues and the challenge of curating sufficiently large training data for each category, limits the effectiveness of classical slot-filling methods in extracting comprehensive user profile information. For instance, consider the sentence: "Obtaining a <u>bachelor's degree</u> was so hard for me, but finally I got it last week. Hooray!". Identifying the phrase "bachelor's degree" as the user's highest level of education allows us to further infer, based on the typical duration of academic programs, that the user is likely at least 21 years old. To overcome these kinds of limitations, we heuristically construct a set of positive and negative examples for two main target concepts defined in our ontology alongside their four sub-concepts related to family roles (e.g., being a mother or husband) and twenty one sub-concepts related to educational statuses (e.g., holding a master's degree or being a second-grade elementary school student). These examples include idiomatic expressions and complex sentence structures to help the model effectively distinguish genuine occurrences of a concept from deceptive cases. We then prompt the LLM using these examples as in-context information in a few-shot learning setup. As depicted in Fig 2, for

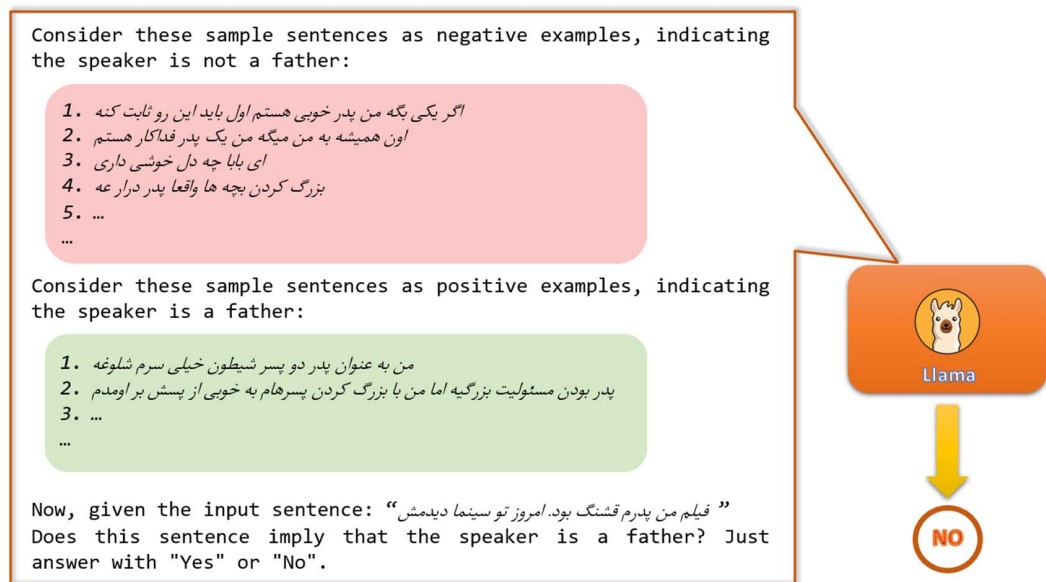

**Fig 2. Few-shot learning prompt used for our LLM-based inference, utilizing examples from the ontology concept 'IsFather'.** Translations of the negative samples are: 1. 'If someone says "I'm a good father," they first have to prove it.', 2. 'He always tells me I'm a devoted father.', 3. 'Oh man, you're so optimistic' (a colloquial expression in Persian with 'dady' word), and 4. 'Raising kids is really exhausting.' (a Persian idiom with 'father' word). Positive samples include: 1. 'As the father of two mischievous boys, I'm really busy.', and 2. 'Being a father is a big responsibility, but I've managed it well raising my sons.'. The input sentence to be classified: 'The movie I am father was beautiful. I watched it in the cinema today'.

each concept of interest (e.g., being a father), we provide a few positively and negatively labeled utterances and prompt Llama-3-70B [33], our underlying LLM, to infer the correct label for the given utterance.

**2.1.4. Step 2: Ontology-based information extraction.** An ontology represents a set of concepts and their relationships in a particular domain in which each concept has a unique identifier linked to at least one natural language term. This enables us to capture inherent semantic and ontological information in human language [37–39]. Using ontology, we can map unstructured information extracted in previous steps to the structured concepts which give us two main capabilities: 1. inference of extra information and 2. detecting semantic conflicts between presented information.

We implement the ontology based on OWL2 ontology language [40] and Pellet reasoner [41]. It contains commonsense knowledge about persons, degrees, jobs, and residential places. For persons, it includes information on age, name, marital status, and family titles with their hierarchies such as father, son, etc. For jobs, it classifies coarse job titles associated with relevant workplaces and required educational degrees. The degree category covers education levels from elementary school to doctoral specialty. Residence includes hierarchical residential information from cities to provinces, countries, and geological directions. Therefore, the ontology has six main classes:

- **Person** includes subclasses of 'Child' (with 'Boy' and 'Girl'), 'Parent' (with 'Father' and 'Mother'), and 'Sibling' (with 'Sister' and 'Brother'). It also includes 'Man', 'Woman', and 'Spouse'. 'Man', distinct from 'Woman', consisted of 'Boy', 'Father', 'Brother', and 'Husband', while 'Woman' includes 'Girl', 'Mother', 'Sister', and 'Wife'. The 'Spouse' class denotes married individuals with titles like 'Husband', 'Wife', 'Mother', and 'Father'. Fig 3 shows a simplified depiction "Person" class. It is worth noting that our actual ontology is more sophisticated.

- **Name** has two subclasses: 'First Name' and 'Last Name' for which the 'First Name' includes a gender property that allows us to assign and distinguish common gender-specific names. Assigning instances to each class results in 294 female and 275 male first names.

- **Residence** includes 'Section' (neighborhoods), 'City', 'Province', 'Country', and 'Region' (geological directions). It captures hierarchical information with 13 relationships such as 'hasCity', 'IsInProvince', 'hasCapital', etc. Additionally, we assign instances to this class and its subclasses by adding the list of all Iranian cities, provinces and geological directions of provinces (such as north, south, etc.) alongside the list of the other countries with their capital cities. It ends up to 1,560 cities, 235 countries, 32 provinces and 6 main directions.

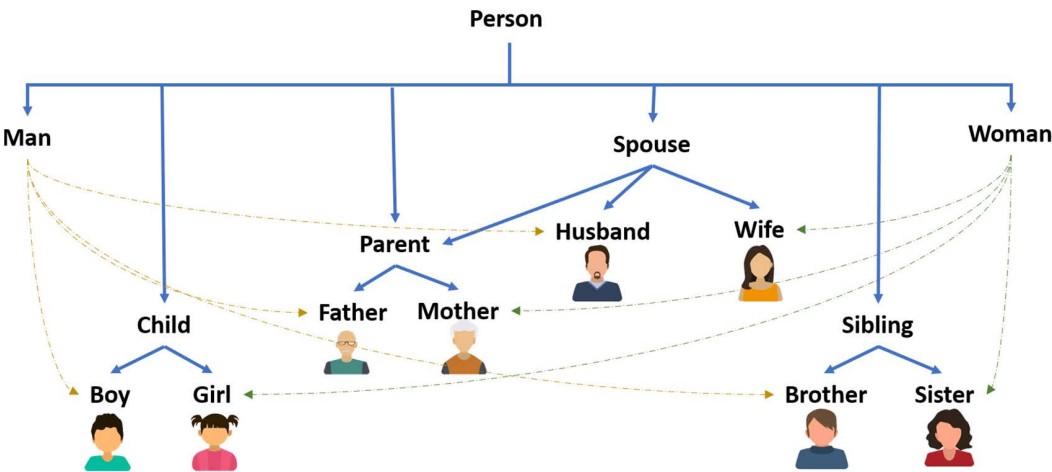

**Fig 3. The hierarchy of the main class of 'Person' in our ontology.**



- **Job** has 18 subclasses (e.g., employee, engineer, doctor), detailing general job information. For instance, doctors require a professional PhD, while engineers need a bachelor's degree. We add 250 job instances, where each one can be assigned to properties such as the type of the job (private, governmental, or freelance), the general field of the occupation (e.g., a service job or a craft), and the minimum degree required.

- **Degree** outlines educational levels with 13 subclasses: elementary school, middle school, high school, pre-university, diploma, associate degree, bachelor's degree, master's degree, PhD, post-doctoral, professional doctorate, specialty, and sub-specialty. Subclasses also capture lower levels (e.g., first to sixth grade). Each level includes a minimum age property that helps us to infer the age range of the user based on their education or the degree required for their job (e.g., minimum age of 7 for a second-grade elementary student or 26 for a pharmacologist).

- **Workplace**, with 81 subclasses, links to the 'Job' through the 'hasWorkingPlace' property and allows inference of job titles (e.g., such as deducing carpenter from working in a carpentry). Overall, the ontology comprises 12 complementary inference rules define through Semantic Web Rule Language (SWRL) [42], 47 class relations, and 29 properties enabling the assignment of data values to instances.

Fig 4 illustrates a simplified overview of the ontology that includes the main concepts, their subclasses, and a subset of the relationships among them. The full ontology is more complex, featuring deeper subclass hierarchies, additional properties associated with each concept, and a richer set of relationships between both main and subordinate classes.

**2.1.5 Inference.** Each ontology concept is associated with the corresponding synonyms or different form of its writing style through which we can later map the extracted information to them. For example, 'cemetery', as a workplace instance, has synonyms of graveyard and burial site. Another example is the country instance of 'ایالات متحده آمریکا' /eja:la:t-e motahhede-je a:mrika:/ (USA) which is accosiated with its different orthographic types such as 'آمریکا' /a:mrika:/ (America) or 'امریکا' /e:mrika:/ (America without diacritic mark as a different writing). Based on this structure, after mapping, ontology

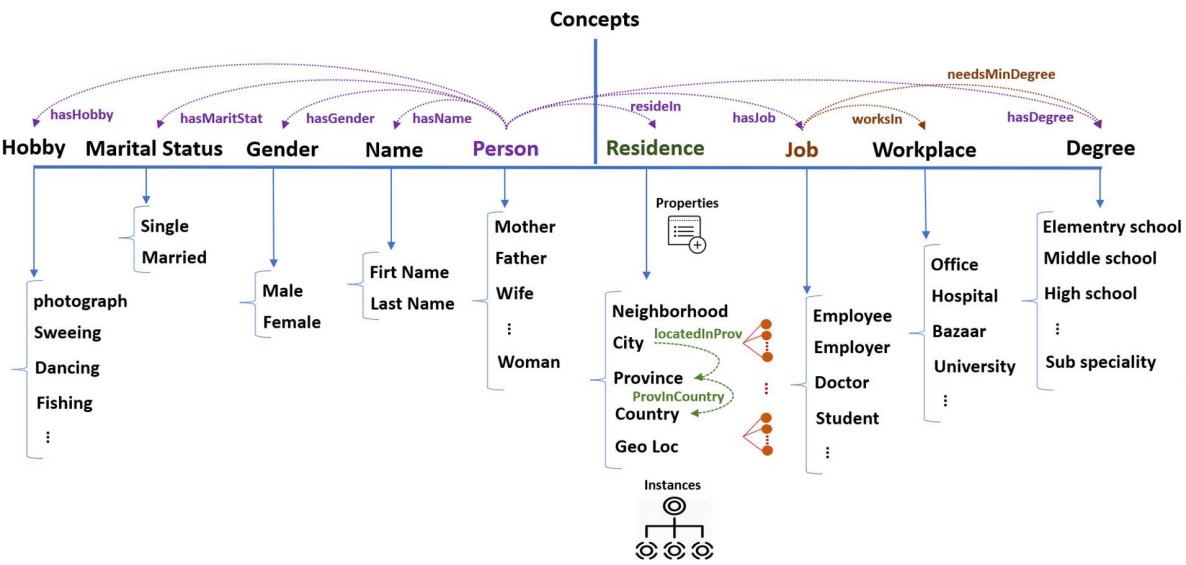

**Fig 4. A simplified view of the concepts in the ontology.** The arrows at the top between concepts represent a subset of inter-class relationships. Each concept also has properties and intra-class relationships, shown by green arrows, as well as its own instances. Additionally, concepts may have deeper internal hierarchies, depicted using red hierarchical structures. The 'Residence' class illustrates these features as an example.

can infer more personal information such as having at least one child, being once married and being female based on the mapped information of being mother.

**2.1.6 Conflict detection.** When we extract user profile information, there is always a chance of getting conflicting information. It is very important to sensitive applications to detect frauds or anomalies. An example of such inconsistency is a person who lives in the city of Tehran (Iran) but has previously mentioned residing in the country of Nigeria. We tailor our ontology to detect pragmatic contradictions or semantic conflicts by detecting any inconsistency between injected information. We categorize these conflicts into the following 11 types reported in Table 1.

At the end of this section, in order to find a better perception of our approach, Fig 5 illustrates the overall workflow where system processes the user input and populates the profile as output.

## 2.2 Ethical consideration

From an ethical perspective, we tried our best to define types of personal attributes that are general without any unique identifier (e.g., social security numbers, addresses, or postal codes). These attributes are not linked to any specific individual and primarily reflect general user preferences (e.g., hobbies). At most, they aim to identify traits associated with particular ages, occupations, or socio-economic situations. This data is a valuable asset in various applications, including personalized dialogue systems, automatic persona detection in persona-grounded chatbots, recommendation systems, and even policy-making through big data analysis for specific age groups or job categories. We are fully aware of the potential misuse of such systems in commercial products. However, it is important to emphasize that our research is purely academic and does not involve directly in collecting user data without explicit or implicit consent.

**Table 1. Types of conflict categories that our method can detect in the user's presented information.**

| Conflict Type | Example | Description |
|---|---|---|
| Geological Direction vs. City/Province | I live in the south of Iran, and to be more specific, I reside in the city of Sari. | Sari is in north direction of Iran not south |
| Country vs. City | I live in Sweden and in the city of Oslo. | Oslo is the capital city of Norway |
| City vs. Province | I live in the city of Goldasht, and in fact, in the province of Gilan | 'Goldasht' city is in 'Isfahan' province |
| Province vs. Country | I am currently living in Iran and living in the state of California | 'California' state is in USA, not Iran |
| Gender-based Name vs. Gender | My name is Mohammad. I am a good-looking girl | Male name of 'Mohammad' for female user |
| Gender vs. Social Titles | Recently, I have become a mother and my wife is not responsible at all. | being a mother is discord with having a wife |
| Gender vs. Job | My name is Kian. I have been out of work for several years and am now officially a housewife | Male name of 'Kian' for person with female job title of 'housewife' |
| Age vs. Job | I am 12 years old and on weekends I work as a specialist surgeon at the hospital | Improbable age-job combinations |
| Age vs. Degree | I just turned 17. I recently got my associate degree | Unusual age-degree combinations |
| Job vs. Degree | I am Sara. I am 22 years old and have a diploma. I am currently working as an optometrist | Insufficient education level for the job title |
| Parenthood vs. Marital status | I have two sweet daughters. Although, I have never married, now looking for a man. | Having children while never getting married |

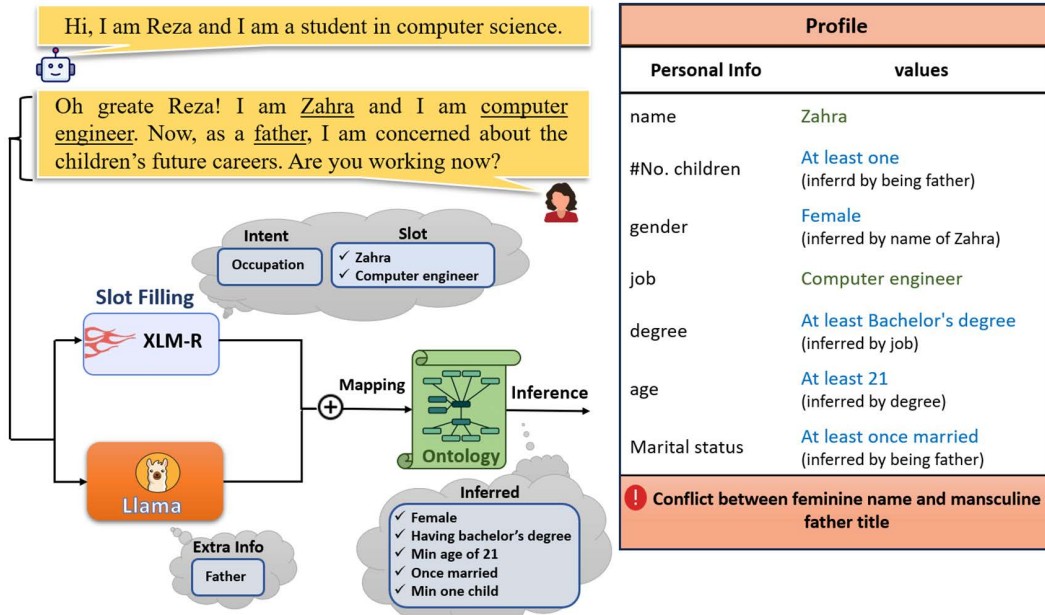

**Fig 5. The overall workflow of the proposed approach.**

## 3 Experimental settings

In this section, we outline the detailed experimental setup in our study, including the dataset and parameter setting for each step.

### 3.1 Dataset

We use the data from Safari and Shamsfard [43]. It is a collection of Persian chit-chat conversations from five main input sources as shown in Table 2. This data is collected from different input sources, including human-human conversations collected through an online persona-grounded chatting site with a gamification method to manually and automatically translated dialogues of ConvAI2 corpus [44] (an extension of PersonaChat corpus [45]). In overall, the dataset contains 6,652 utterances with 169,047 tokens and vocabulary size of 7,660. To the best of our knowledge, this is the only available dataset in Persian that contains chit-chat dialogues annotated with user's personal information. Each utterance is labeled with one to at most three topics (intents) selected from 13 categories. Tables 3–5 show information about topic distribution, frequency of extracted personal information from each utterance, and distribution of annotated slot tags respectively. We divide the dataset into 5,217 utterances for the training, 706 for validation, and the rest of 729 utterances for the test set. To ensure a balanced distribution, we allocate 80% of each intent class and each slot type to the training and 10% is dedicated to each validation and test set.

### 3.2 Slot filling

We inspect and compare several different encoder-only language models with our slot filling (SF) and intent detection (ID) multi-task learning. Specifically, we use *JointBert* [31], based on the BERT transformer model [46], *Stack Propagation* [47], based on traditional RNN architecture that uses joint training with a hierarchical structure, and *MISCA* [36], a transformer-based framework designed for multi-intent detection. We train them all with a maximum length of 110 tokens and

**Table 2. Five main input sources of data collected in the research of Safari and Shamsfard [43].**

| Type | Resource | Dialogue No. | Utterance No. |
|---|---|---|---|
| General | English training sites with Persian transcriptions | 74 | 571 |
| Data Collection | A dataset of free discussion dialogues | 99 | 2918 |
| Translation | Manual translation of ConvAI2 corpus [44] | 72 | 1000 |
| Gamification | Gamification through an online chatting site with random profiles for participants | 94 | 1557 |
| Augmentation | Augmenting semantic frames in the previously collected data by semi & fully automatic methods | – | 336 |
| Generation | Instructing GPT-3.5 to generate dialogues without/with desired personal info. | 335 | 5094 |

**Table 3. Distribution of topics (intents) in both single and multi-intent mode in the dataset.**

| Topic | Single intent | Multi intent | Topic | Single intent | Multi intent |
|---|---|---|---|---|---|
| Greeting | 283 | 288 | Residence | 614 | 717 |
| Weather | 48 | 75 | Marriage | 91 | 132 |
| Age | 185 | 230 | Family | 604 | 741 |
| Education | 155 | 206 | Hobby | 1,808 | 1,966 |
| Gender | 65 | 103 | Name | 466 | 487 |
| Occupation | 998 | 1,173 | Other | 1,182 | 1,190 |
| Goodbye | 153 | 162 | | | |

**Table 4. Distribution of extracted personal information in annotated data.**

| Marital Status | 85 | Age | 113 |
|---|---|---|---|
| Occupation | 518 | Name | 385 |
| Gender | 52 | Residence | 390 |
| No. Siblings | 293 | No. Children | 185 |
| Hobby | 1,282 | | |

**Table 5. Distribution of slot tagset in annotated data.**

| Slot Tag | # | Slot Tag | # | Slot Tag | # | Slot Tag | # |
|---|---|---|---|---|---|---|---|
| B-Name | 385 | I-Name | 11 | B-Residence | 390 | I-Residence | 43 |
| B-Age | 113 | I-Age | 118 | B-Hobby | 1282 | I-Hobby | 866 |
| B-Child | 185 | I-Child | 231 | B-Job | 518 | I-Job | 224 |
| B-Sibling | 293 | I-Sibling | 362 | B-Marriage | 85 | I-Marriage | 32 |
| B-Gender | 52 | I-Gender | 1 | O | | 81,594 | |

a batch size of 64 in the single-intent mode (considering only the dominant topic for utterances with multiple intents). In order to identify the best trained model in JointBert, we add a new criterion by summing up intent accuracy and slot-F1. Based on this criterion, the best version is saved at the end of the training session and loaded in the test phase. Also, in MISCA framework, for word embedding we utilize the Persian Roberta v3 (from HooshvareLab) as the underlying Persian LM.

### 3.3 Large language models

We also investigate the performance of LLMs during our experiments. For this purpose, similar to previous work [48–50], we use a question answering framework for Llama-3-70B (Llama-70b-chat-hf) and GPT-4o with some differences in our

prompt engineering approach. Usually through these prompts, the model is asked to do slot filling and intent detection, and the prompt contains descriptive slot names alongside the sample of their accepted values while they ask for a specific output schema. However, in our prompt, we directly ask about each of the nine personal information categories, labeled as slot types in the dataset, namely: gender, marital status, age, name, occupation, residence, hobby, number of children, and number of siblings (e.g., "Is the speaker male or female?" or "What is the speaker's name?"). We also include the previous chatbot utterance from dialogue history to improve the prompt accuracy: such as, 'Chatbot: where are you from? User: 'Shush. Have you heard anything about the Burnt City?' (Shush is one of Iranian historical cities). Additionally, we ask the model to work in a multi-intent mode by selecting the main topics among the list of accepted values (to find the prompt refer to S1 File).

It is worth noting that we use English for prompting the LLMs while keeping the context and input utterances in Persian to isolate the impact of input language on performance. As another reason, some prior studies [51,52] show using English prompts results better for multilingual LLMs compared to local languages and also, employing Persian prompt with translation of key terms like "intent" may bring ambiguity or errors into the model.

## 4 Results

As mentioned in earlier sections, our final model is capable of both extracting user profile information and identifying semantic conflicts (inconsistencies). In this section, we present the results for each of these two tasks.

### 4.1 User profile extraction

Table 6 shows the performance of our multi-step approach in extracting user profile compared to GPT-4o and Llama-3-70B in a zero-shot manner. The detail of each prompt is shown in S1 File). We report the F1-score, precision and recall for each model in Table 6. We observe that our method significantly outperforms both LLMs, achieving 17.08% and 24.51% improvement in profile-F1. GPT-4o ranks second, outperforming Llama with a 7.43% increase in F1. To assess the statistical significance of these improvements, we also perform paired *t*-tests over the five folds. The results confirm the superiority of our method over both Llama-3-70B ($t(4) = 39.10$, $p < 0.001$) and GPT-4o ($t(4) = 29.50$, $p < 0.001$). To gain a clearer unders*t*anding of performance across individual user attributes, we report the results by information type in Table 7.

The results demonstrate the superiority of our approach in extracting all attributes except Name. GPT and Llama exhibit significant underperformance in extracting Marital Status, Gender, and Age, primarily due to their limitations in inferring values from explicit contextual information. For instance, they struggle to deduce age based on typical age ranges for specific job titles or the user's highest degree. While GPT can infer gender from certain gender-specific names, it still falls short in many cases, including the common example of 'Mohammad'. Additionally, GPT avoids making marital status inferences based on parenthood or references to having children, whereas Llama is more inclined to infer both marriage and gender. However, Llama also introduces biases in assigning these attributes. In Section 6.1, we conduct a detailed analysis of these cases.

**Table 6. Performance comparison of our multi-step approach and zero-shot LLMs in user profile extraction.**

|              | Profile-F1 | Profile-Precision | Profile-Recall |
|--------------|------------|-------------------|----------------|
| Our Approach | **90.46**  | **91.07**         | **89.86**      |
| GPT-4o       | 73.38      | 69.72             | 77.44          |
| Llama-3-70B  | 65.95      | 55.85             | 80.51          |

**Table 7. The performance of LLMs and our method across different personal attributes in terms of F1, Precision (P) and Recall (R).**

|  | Our approach | | | GPT-4o | | | Llama-3-70B | | |
|---|---|---|---|---|---|---|---|---|---|
|  | F1 | P | R | F1 | P | R | F1 | P | R |
| Marital Status | **81.25** | 72.22 | 92.85 | 57.14 | 85.71 | 42.85 | 69.23 | 75.00 | 64.28 |
| Gender | **87.23** | 95.34 | 80.39 | 64.10 | 92.59 | 49.01 | 78.18 | 72.88 | 84.31 |
| Age | **84.93** | 93.93 | 77.50 | 30.43 | 100 | 17.94 | 29.16 | 87.50 | 17.50 |
| Name | 90.47 | 92.68 | 88.37 | **95.45** | 93.33 | 97.67 | 88.88 | 85.10 | 93.02 |
| Job | **99.08** | 100 | 98.18 | 92.59 | 90.90 | 94.33 | 71.75 | 60.25 | 88.67 |
| Residence | **94.94** | 100 | 90.38 | 89.47 | 82.25 | 98.07 | 65.80 | 49.51 | 98.07 |
| Hobby | **94.65** | 100 | 89.84 | 80.48 | 83.89 | 77.34 | 81.10 | 81.74 | 80.46 |
| #Siblings | **97.36** | 100 | 94.87 | 89.32 | 82.14 | 97.87 | 70.14 | 54.02 | 100 |
| #Children | **100** | 100 | 100 | 76.54 | 65.95 | 91.17 | 62.50 | 54.34 | 73.52 |

## 4.2 Conflict detection

We assess the performance of our approach from the aspect of conflict detection which also involves using the full pipeline. We compare it against four high-performance LLMs: Llama-3-70b, GPT-3.5-turbo, GPT-4 and GPT-4o. As a result, we collect a set of 100 utterances including 80 conflictive and 20 non-conflictive statements where for each 11 conflict categories, nearly 8 cases with contradiction and 2 cases without any semantic discord are included. This set is exposed to LLMs through prompting. We prompt the model to identify any probable implicit or explicit semantic contradiction with explanation or state none if there is not any conflict (to find the prompt refer to S1 File). We manually inspect the results of all models and report the performance in Table 8 in terms of accuracy.

Our method achieves the superior performance with overall accuracy of 92%. However, GPT-4o, as the top-performing LLM model, is 21% less accurate in overall. GPT-3.5-turbo performs the worst, with an accuracy of 32%, while Llama-3-70b is superior than GPT-3.5 by 13%, achieving 45% overall accuracy. GPT-4 is next, performing 15% better than Llama-3-70b but still 5% lower than GPT-4o.

We also inspect accuracy for conflict and non-conflictive utterances reported in the second and third rows of the table. This analysis reveals that Llama-3-70b and GPT-4o, with lower non-conflictive accuracy and higher conflicitve accuracy compared to GPT-3.5-turbo and GPT-4, are more inclined to label conflicts, even at the cost of precision. Our approach is also incline to pass cases as non-conflictive due to missing extraction of some explicit information and this error propagates through the pipeline.

Fig 6 shows the accuracy of each model. As the results show, our approach is superior across all categories except for detecting country vs. city which relates to missing information in the first step. LLMs have the worst results in three categories of city vs. province, gender vs. social titles and gender vs. job which is due to not recognizing the values in some cases or not being able to correctly infer conflictive information from explicit values. We present a detailed analysis of these results in §6.1.1.

**Table 8. Performance of our approach on conflict detection compared to high-performance LLMs.**

|  | Ours | Llama-3-70b | GPT-3.5 | GPT-4 | GPT-4o |
|---|---|---|---|---|---|
| Overall accuracy | **92** | 45 | 32 | 60 | 71 |
| Conflict accuracy | **91** | 51 | 33 | 50 | 66 |
| Non-conflictive accuracy | **95** | 19 | 23 | 95 | 85 |

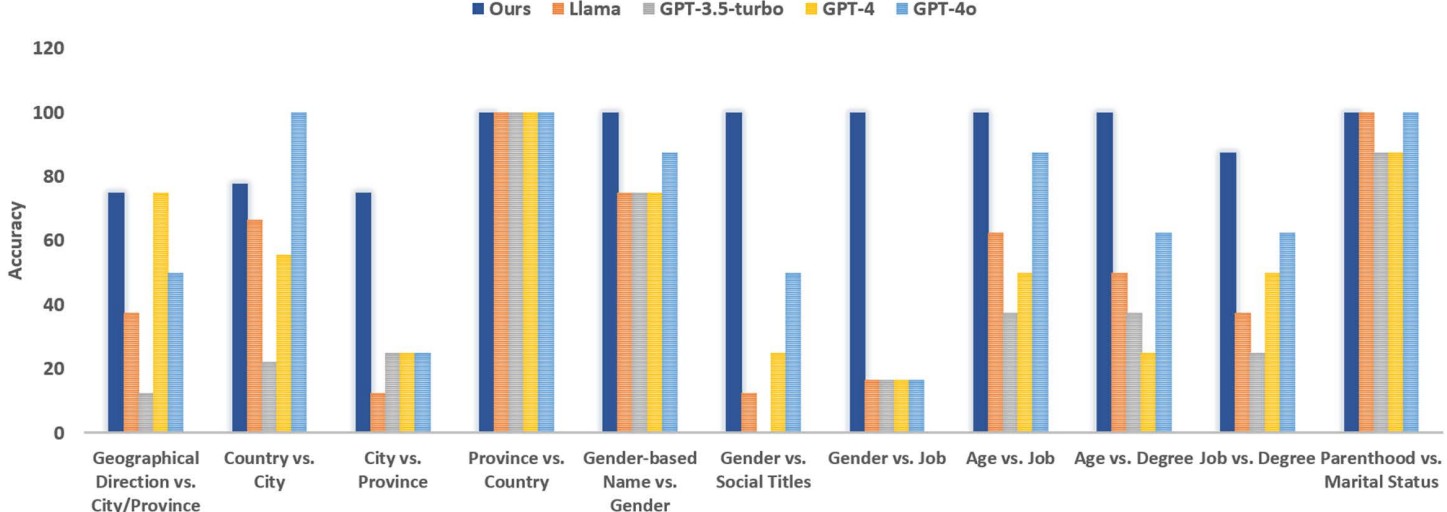

**Fig 6. Visualized accuracy comparison between our multi-step approach and LLMs across different conflict categories.** Corresponding numerical values are reported in S1 Table in S1 File.

## 5 Analysis

In this section, we analyze the performance of our approach from multiple aspects. We focus on evaluating different configurations and enhancements for LLMs to determine whether they can achieve comparable performance to our multi-step method. Particular attention is given to the slot filling, which we identify as the bottleneck of performance in our pipeline.

### 5.1 Performance of slot filling compared to base methods

Initially, we compare the performance of our slot filling method that leverages XLM-Roberta with base techniques described in §3.2. The results are reported in Table 9. We perform Intent Detection (ID), the auxiliary task in multi-task learning, in both single and multi-intent modes for all methods, except for Stack Propagation [47], where we skip multi-intent due to its relatively low performance in the single mode. The last two rows of the table related to LLM results.

According to Table 9, our SF method outperforms all base approaches. Compared to JointBert, the next best-performing model, it achieves 4.12% increase in intent accuracy (single-intent mode) and nearly 5% improvement in slot-F1. However, in the multi-intent setting, our model's performance drops 7% in intent accuracy and 1% in slot-F1.

**Table 9. Comparing the performance of our slot filling methods against base methods.**

| Methodology | Intent Accuracy | Slot-F1 | Slot-Precision | Slot-Recall |
|---|---|---|---|---|
| Our Method (single-intent) | **91.22** | **88.07** | **89.04** | 87.13 |
| Our Method (multi-intent) | 84.21 | 86.87 | 85.73 | 84.63 |
| JointBert (single-intent) [31] | 87.10 | 83.19 | 84.10 | 82.30 |
| JointBert (multi-intent) | 80.38 | 82.10 | 85.05 | 79.35 |
| Stack Propagation [47] | 75.30 | 77.00 | 73.19 | 81.25 |
| MISCA (single-intent) [36] | 81.75 | 79.10 | 82.31 | 76.13 |
| MISCA (multi-intent) | 48.42 | 78.03 | 79.44 | 76.67 |
| Llama-3-70b (multi-intent) [33] | 40.87 | 70.76 | 56.67 | 94.19 |
| GPT-4o (multi-intent) [32] | 49.24 | 73.45 | 58.51 | **98.64** |

Additionally, comparing JointBert to Stack-Propagation highlights the advantages of using a pretrained language model. The MISCA framework with its more complex structure, requires significantly more training epochs (30 for JointBert vs. 160 for MISCA) and is more inclined to output multi-label intents. Even in multi-intent mode, its performance remains lower than JointBert with 32% drop in intent accuracy and 4.07% decline in slot-F1.

For the evaluation of the two LLMs in the last rows, we automatically calculate the scores and also revise the outputs. The results show that GPT-4o outperforms Llama with 2.69% increase in slot-F1 and 8.37% increase in intent accuracy. However, both models underperform compared to our method in multi-intent mode, with slot-F1 in GPT-4o and Llama being 13.42% and 16.11% lower, respectively. Notably, compared to the multi-intent version of MISCA, the intent accuracy of GPT-4o is 0.82% better while works in a zero-shot scenario. Additionally, high slot recall and low precision in the LLMs indicate their tendency to take any probable information as slot values, even at the cost of reducing accuracy.

## 5.2 Structure and feature selection for slot filling

Given the nature of the dataset, some utterances contain multiple topics or intents and it makes slot types non-exclusive to a single intent. To address this, we also explore disjoint training of ID and SF during 30 epochs, using a batch size of 8 and a maximum sequence length of 110 tokens. As shown in the second row of Table 10, joint training proves more effective, as separate training results in 1.37% decrease in intent accuracy and 1.61% drop in slot-F1.

Next, we examine the impact of incorporating the previous utterance as an input feature. This proves beneficial when the current intent aligns with the prior topic or when personal information is a response to an earlier utterance. The results, shown in the last row of table, demonstrate 1.78% increase in intent accuracy and 0.83% improvement in slot-F1.

## 5.3 Language model comparison for slot filling

Moreover, we evaluate the performance of our joint slot filling method using different pretrained Language Models (LMs), training each for 30 epochs with a batch size of 64 and a max token length of 110. The results are shown in Table 11. The 'Sentence Acc.' column represents sentence accuracy and indicates the percentage of utterances where both intent and slots are correctly detected. In this table, Bert, Roberta, Albert, and DistilBert represent Persian pretrained LMs from Hooshvar Lab [53], while XLMRoberta is a multilingual model from Facebook AI [35]. FaBert [54] is a recently released

**Table 10. Comparing the slot filling performance while disjointly training ID & SF tasks, joint training tasks and injecting the previous utterance as input feature.**

|  | Intent Accuracy | Slot-F1 | Slot-Precision | Slot-Recall |
|---|---|---|---|---|
| ID + SF | 89.85 | 86.46 | 86.64 | 86.33 |
| Joint Training | 89.44 | 87.24 | **89.29** | 85.26 |
| Joint Training +prev utter | **91.22** | **88.07** | 89.04 | **87.13** |

**Table 11. Comparing the performance of different language models as the basis of our joint training method.**

| Model | Intent Accuracy | Slot-F1 | Slot-Precision | Slot-Recall | Sentece Accu. |
|---|---|---|---|---|---|
| BERT | 89.84 | 85.06 | 84.61 | 85.52 | 80.52 |
| FaBERT | 70.91 | 62.88 | 66.66 | 59.51 | 54.04 |
| Roberta | 87.10 | 83.19 | 84.10 | 82.30 | 75.99 |
| AlBERT | 88.75 | 84.80 | 87.46 | 82.30 | 79.28 |
| DistilBERT | 90.12 | 84.09 | 84.55 | 83.64 | 79.83 |
| XLM-Roberta | **91.22** | **88.07** | **89.04** | **87.13** | **83.40** |

model exclusively pretrained on Persian language using both formal and informal texts. According to the results, XLMRoberta performed the best, with a 1.38% increase in intent accuracy, 3% increase in slot-F1, and 2.88% increase in sentence accuracy compared to Bert.

## 5.4 Prompt engineering and fine-tuning LLMs on slot filling

Since the zero-shot performance of LLMs on the slot filling, reported in the last two rows of Table 9, was significantly low in terms of precision and slot-F1, we conduct an analytical study presented in Section 6.2. Based on this analysis, we engineer the prompt again and refine it by incorporating complementary instructions and addressing common errors. For example, we guide the model to distinguish between the user's workplace and residence. This helps prevent mistakes such as misinterpreting "Sa'dabad Palace" as the user's residential place in the utterance of "I am the museum guide of Sa'dabad Palace".

Additionally, we define the Persian expressions of "تک فرزند بودن" /tæk færzænd budæn/ and "تک بچه بودن" /tæk bætʃɛ budæn/ (being an only child) as the most persistent sources of errors in LLMs to ensure that they correctly assign zero to the number of siblings. We also clarify the definition of "hobby" as an activity that can be performed during leisure time to prevent errors such as misinterpreting user's interest in "tasting Indian food" as their hobby in the utterance of "I am fond of tasting Indian foods."

To improve consistency in generated outputs and align them with gold values while reducing manual verification effort, we also establish a structured output template. For instance, marital status should be strictly limited to either "married" or "single." Similarly, the number of children should be formatted as "#No daughters, #No boys, #No children" to ensure precise representation. For multi-value slots, such as the user's name, job, or hobby, we instruct the model to treat them as multi-valued by separating entries with commas. As another refinement, to address the issue of incorrect age calculation in LLMs, we instruct the model to output the year of birth if mentioned, rather than performing the calculation itself. Then, as a post-processing step, we calculate the age and align it with the gold-standard age to avoid such erroneous cases.

Furthermore, we explicitly direct the model to avoid inferring gender based on names, because this can introduce bias in LLMs. Additionally, gender inference is not part of the explicit information extraction in first step. Previously, we manually checked the results and with leniency accepted correct values. Therefore, excluding it reduces the manual effort to verify results. (to find refined prompt, refer to the S1 File).

On the other hand, to further refine the generated outputs of LLMs and align them more easily with gold values, we implement and apply a post-processing step to both the gold and predicted values. This process includes a set of regular expressions for each slot type to standardize the format (such as standardizing the value of "چهار گل دختر" /tʃeha:r gʊl doxtær/ (four lovely daughters) to "4 دختر" 4 /doxtær/ (4 daughters) for slot type of the number of children). Also, for slot types with nearly closed sets of values, such as name or residence slots, we compile a list of frequent male/female names and the common city names with the list of all countries. This allows us to compare values and remove Persian suffixes that function similarly to English demonyms in residential contexts (such as "ی" (i-) in "رشتی" /ræʃti/ ("residing in Rasht") in the utterance ".من رشتی هستم" /mæn ræʃti hæstæm/ ("I am from Rasht")) and also enable us to address clitic pronouns that attach directly to names and replace the verb' "to be" (such as "م" in "من مهسام" /mæn mæhsa:-m/ ("I am Mahsa")).

Regarding all of these refinements, we repeat the slot filling task on GPT-4o as the best resulted LLM in the previous experiment. The results are shown in the third row of Table 12.

After improving the prompt, comparison of GPT's performance shows that precision has increased by 4.66%. It indicates the effectiveness of our complementary instructs; however, the recall shows nearly 10% decrease and therefore, F1 score remains almost the same. Manual investigation of results reveals that after refining, model answers more concise and appropriate specially for 'hobby' slot (such as generating 'traveling' instead of 'traveling to cities with history and culture'/'traveling and exploring new places' in previous version or 'reading books' instead of 'spending time by reading books'). Also, defining hobby helped model to ignore interests or irrelevant values (such as ignoring 'liking Chinese food' or 'preserving the environment'). The model



**Table 12. Comparing the performance of our slot filling method with LLMs in zero-shot learning with initial/refined prompt and also fine-tuned version of Phi-4.**

| Model | Precision | Recall | Slot-F1 | Intent Acc. |
|---|---|---|---|---|
| Our method (trained) | **89.04** | 87.13 | **88.07** | **91.22** |
| GPT-4o (zero-initial) | 58.51 | **98.64** | 73.45 | 62.41 |
| GPT-4o (zero-refined) | 63.17 | 88.70 | 73.79 | 77.77 |
| Llama-3.1-8B (zero-refined) | 25.77 | 67.44 | 37.29 | 71.19 |
| Phi-4 (zero-refined) | 40.37 | 75.88 | 52.70 | 72.83 |
| Phi-4 (fine tuned) | 58.67 | 89.57 | 70.90 | – |

'zero' beside model name indicate using zero-shot technique with initial prompt ('initial') or refined prompt ('refined').

also now adheres more strictly to generating 'I don't know' when no information is available, instead of responses like 'the residential city is not specified' or 'no specific job has been mentioned' from the initial version. More importantly, the refined version significantly reduces the extraction of information from previous chatbot utterance which is a common source of error in LLMs. It leads to a notable improvement in precision. Furthermore, the model's more concise responses in the refined version prevent it from generating overly expressive or generalized terms for the 'job' slot. For example, it now avoids responses like 'works in an IT company' for utterances such as 'I work in an IT company'. Since the gold data does not accept general descriptions and requires precise job titles, in the previous experiment with initial version, we tolerated such responses as correct.

In the next step, we proceed to fine-tune LLMs locally. We want to assess how much we can enhance their performance with our limited training data, relative to the large number of model parameters. As fine-tuning candidates, we need LLMs that are both small enough to be trained locally and capable of delivering reasonable results compared to large models like GPT-4o. Since the Llama family demonstrates performance on par with GPT, we select the latest released model, Llama-3.1-8B, along with Phi-4 [34], which was recently published with 14B parameters and has shown strong performance in reasoning tasks. The fourth and fifth rows of Table 12 present their zero-shot performance with the refined prompt. Phi-4 outperforms Llama-3.1-8B significantly, with a 15.41% higher slot-F1 score. Consequently, we fine-tune only Phi-4 on our dataset for 2 epochs using the LoRA [55] technique (r = 16). The results are shown in the last row of Table 12. Fine-tuning substantially improves Phi-4's performance, with an 18.2% increase in slot-F1, 18.3% increase in precision, and 13.69% increase in recall. However, it still lags behind our slot filling technique, with 17.17% lower slot-F1 score. This gap may be due to the limited size of our dataset, which is insufficient to fully optimize the model's behavior given its large number of parameters. In Section 6.3, we provide a detailed analysis of LLM performance with fine-tuned settings. In general, from the results of the table, we conclude that LLMs with high recall are more intended to take any acceptable value for personal attributes and it comes at the cost of lower precision.

Finally, to evaluate the impact of refining instructions related to the auxiliary task of intent detection, we measure the models accuracy in single-topic mode. For LLMs, we consider a prediction correct if at least one of the mentioned topics (intents) matches any of the gold topics. Although refinement improves GPT-4o's performance by 15.36%, it remains 13.45% lower than our joint SF and ID model.

# 6 Discussion

In this section, we describe our analytical studies. The first two subsections present analysis on results related to the overall performance while the other sections inspect results on each aspect of the process.

## 6.1 Analysis of profile extraction results

Our investigation from our multi-step approach reveals that our model lags in detecting information decoded contextually or explicitly in the previous utterance. For instance, in the dialogue: "-What do you do dear Zahra? - I am college student.",

model do not register name of the user explicitly mentioned in the previous utterance or "-I am interested in jogging. What about you? -Well, it seems that we both share the same interests." where model is lame in extracting the more contextualized hobby information decoded in both previous and current utterances. Although LLMs are more aware in these cases but they still have problem with subtle articulated samples such as "-Are you one of the two daughters in your family? -Yes, and also I have a son and a daughter myself." from which we can infer that user has one sister but LLMs miss this information.

On the other hand, LLMs usually take applicable values in a user's question as their personal information. For example, in the utterance "Yes, the weather here is warm. Is weather in Tehran cold?", models take 'Tehran' as the user's residential city. Also, they tend to take values from previous chatbot utterance which is more common in Llama compared to GPT such as inferring 'cooking' as user's hobby by GPT model in this dialogue: 'Chatbot: I personally like cooking more. What name can I call you by? User: I am Ainaz.' Moreover, both models have difficulty in extracting the explicitly stated number of siblings or children. Specifically, they struggle to correctly infer the Persian phrase 'تک بچه بودن' /tæk bætʃε budæn/ (being only child) which is interpreted as having neither any children nor any siblings in GPT and having one sibling in Llama. These three sources of errors are repeated in the other complementary experiments in the following sections.

Inspecting the performance of each LLM shows that GPT-4o lags in extracting some of explicitly mentioned hobbies while in the other zero-shot experiments explained in Sections 5.1 and 5.4, it shows better performance on this attribute. For example, it ignores values in the following utterances: 'I am fond of traveling', 'I am interested in listening music. Do you like it too?' or 'I love music'. Model also miss calculating age in the case of 'I was born in 1375' which is correctly calculated by our method and also by GPT-4o itself in the other upcoming experiments. About implicit extraction, GPT is capable of inferring gender in some cases for gender specific names. Model infers cases such as 'Female' for less common feminine name of 'Ainaz' but it miss 'Male' for the most recognized masculine name of 'Mohammad'. About inferring marital status from being parent or claiming to have children, model impedes from any further inference which is also true for deducing age based on user's highest degree or some specific job titles (such as 'Engineer' which implies at least 23 years old for user to obtain degree associated with this title or 'university student' which implies at least 18 years old). Apart from these general trends, we can also find minor shortages such as ignoring the inference of residence from contextually mentioned value of 'Iran' in this case: 'I have mostly traveled within my country, enjoying the tourist attractions of Iran' which is correctly inferred by Llama.

For Llama model, it shows more conspicuous errors, even assigning irrelevant value to an attribute such as following cases: 'I think that I have seen you in the university' (residence = 'university'), 'I am fine. I am from Bafq (a city located in Iran). Where are you from?' (job = 'Bafq'), 'I love the freshness and coolness of the air in summer' (hobby = 'freshness and coolness of the air' while in prompt, we have defined hobby as an activity that user can do in their leisure time) or 'I also love traveling; my last trip was to Turkey' (residence = 'Turkey'). Another typical trend in Llama is assigning general values as job title to the occupation attribute such as assigning 'working in an IT company' in the utterance 'I work at an IT company'. We take these cases as correct with leniency. About inference capability of Llama, model can infer gender and marital status based on the explicit information. However, model shows strong indications of bias in gender fields. For example, it infers 'Male' gender from being Electrical Engineer, Civil Engineer, being interested in football, beating children by the user or even more generic patterns such as probably having sons in the utterance 'I enjoy going to parties. Sometimes, I also take my sons to my friends' houses'. There is also inherently wrong cases such as inferring 'Male' from the most common feminine name of 'Sarah'. For age inference, this model is also incapable in all cases.

As a summary, Table 13 presents a set of case studies comparing the performance of different models in extracting personal attributes. The first column contains the dialogue turn, including both the previous and current utterances. The second column lists the ground-truth personal attributes extracted from the current utterance and the remaining columns show the output of each model.

**Table 13. Case study comparison of personal attribute (profile) extraction between our approach and LLMs.**

| Dialogue Turn | Gold Personal Attributes | Our Approach | GPT-4o | LLaMA-3-70B |
|---|---|---|---|---|
| – Are you a painter or graphic designer, dear <u>Sobhan</u>? – How interesting, you guessed it. Yes, I am actually a <u>graphic designer</u>. | *name* = Sobhan, *gender* = male, *job* = graphic designer, *age$_{min}$* = 22 | *name* = ✗ (missed), *gender* = ✗ (missed), *job* = ✓, *age$_{min}$* = ✓ | *name* = ✓, *gender* = ✓, *job* = ✓, *age$_{min}$* = ✗ (missed) | *name* = ✓, *gender* = ✓, *job* = ✓, *age$_{min}$* = ✗ (missed) |
| – Kids bring so much happiness; I hope you experience it too. – Really? How interesting! What times of the year do you like best for living in <u>Shahrud</u>? | No attribute | No attribute ✓ | *residence* = Shahrud ✗ | *residence* = Shahrud ✗ |
| – Yes, I'm blessed with <u>one son</u> and <u>five daughters</u>. – You've got so many more kids than my family! I actually always wanted a sister, but I'm the <u>only child</u>. | *#siblings* = 0 | *#siblings* = ✓ | *#siblings* = ✓, *#children* = ✗ (0) | *#siblings* = ✓, *gender* = ✗ (male) |
| – No, I'm still <u>single</u> and don't have any children. How many kids do you have? – As I just told you, I have <u>three children</u>, <u>one son</u> and <u>two daughters</u>. Thanks for the conversation. | *#children* = 1$_{son}$, 2$_{daughters}$, *marital_stat* = once married | *#children* = ✓, *marital_stat* = ✓ | *#children* = ✓, *marital_stat* = ✗ (missed) | *#children* = ✓, *marital_stat* = ✓, *#siblings* = ✗ (1$_{brother}$, 2$_{sisters}$) |
| – I find <u>reading historical books</u> and <u>watching romantic movies</u> fascinating. Do you share similar interests? – I think such interests are wonderful too. Do you enjoy <u>music</u>? | *hobby* = reading historical books, watching romantic movies | *hobby* = ✗ (missed) | *hobby* = ✓ | *hobby* = ✗ (music) |
| – My birthday's in Bahman (January). What's your birth year? – I was born in <u>1374</u> (1996) & really enjoy sharing my life experiences with others. | *age* = 29 | *age* = ✓ | *age* = ✗ (missed) | *age* = ✗ (missed) |

In the cell contents, ✓ indicates correctly extracted information and ✗ denotes incorrect (extra or missed) values.

**6.1.1 Analysis of conflict detection results.** Our manual investigation of the results reveals that our approach missed a total of 7 conflictive cases due to not being able to detect them during slot filling process. Two cases involve conflicts between direction and city, where the SF task fails to detect directions. It appears that when they are accompanied by complementary words, the model is more sensitive to find them (e.g., 'north of <u>Iran</u>'/'north of <u>country</u>' vs. 'north').

Additionally, our slot filling method sometimes fails to detect certain capital cities or countries, highlighting the necessity of increasing the diversity of this slot type or attribute in the training set. This results in 2 errors in city vs. province and 2 errors in city vs. country conflicts. Furthermore, information related to certain expressions of residence is more likely to be ignored by our model, although changing the writing style resolved the issue. This indicates a need to add more diversity in expressing the residence slot type in the training set. There was also one error in detecting the conflict between job vs. degree, again related to not detecting the job title.

We also encounter one incorrect case in the non-conflictive category:

"من سجاد هستم. من مادرم مثل همه خالههام پزشک هست. خودم هم مدرک دکترای حرفهای دارم و الان پزشک هستم."

/mæn sædʒɑːd hæstæm/. /mæn mɑːdæræm mesle hæme-je xɑːlehɑːm pezeʃk hæst/. /xodæm hæm mædræk-e doktorɑː-j-e herfe-ʔ-i dɑːræm væ ælɑːn pezeʃk hæstæm/ (My name is Sajad. My mother like all my aunts is doctor. I also have a professional doctorate degree and currently I am a doctor.) The extra 'من' /mæn/ (I), as an emphasis, led the underlying LLM in first step of our approach to wrongly tag the user as a mother, triggering an alarm about the contradiction between the masculine name of 'Sajad' and the feminine title of 'mother'. This indicates the necessity of adding these adversarial samples to the concepts of our ontology in the future work.

Regarding the performance of the other LLMs (GPTs and Llama-3-70b) with which we compare our approach shows that they all successfully pass the conflict tests between province and country. However, detecting conflicts between job

and gender proved to be the most challenging task for them, with only one detected case out of 5 input cases. In all samples of this category, the gender should be implied by the name or gender-based titles (e.g., reference to the user's wife), and only in one case the male gender is explicitly indicated. In this instance, Llama and GPT-3.5 identified it as the main contradiction, while GPT-4 marked it as an implicit conflict with traditions in some contexts. Although GPT-4o did not recognize any conflict in this case, in another case where the user referenced to his wife while being a housewife, the model identified it as the main mismatch.

Detecting contradictions between city and province was also challenging, with only 2 correct answers out of 8 input samples for all models and 1 out of 8 for GPT-3.5. This may be due to the LLMs not recognizing some cities and their insufficient representation in the model's training dataset. In the other situations, LLMs interpret certain city names as typos and attempt to correct them by substituting similar-sounding city names, such as interpreting 'Abdanan' as 'Abadan' or 'Goldasht' as 'Golshahr'.

We also inspect the erroneous cases in gender vs. social titles. In this category, none of the LLMs could detect conflict of this case: 'I am an independent woman. Although I am married, it doesn't taken away my independence. My wife understands me very well.' however, when we replace 'woman' with 'man' and 'wife' with 'husband', GPT-4 and GPT-4o can easily detect the conflict while even with simplifying sentence into 'I am a fortunate man. My husband understands me very well.', GPT-3.5 and Llama still find false contradictions like this one by Llama as another type of model's bias: 'a man, describing their husband as very understanding, which might be unexpected in a traditional Persian culture, where men are often stereotyped as being less emotionally expressive or understanding'.

Another showcase in this category involves investigating the source of error in this case: 'My name is Hadis. I have been married and have a good life with my wife, but we still don't have any children.'. When we replace 'Hadis', a girlish name, with another well-known female name of 'Zahra', the error still remains and models focus on the explicit conflict indicated by 'but' and interpreting it as 'The contradiction between the happy marriage and the absence of children'. When we simplify it into 'My name is Zahra. I have been married and have a very good life with my wife.', GPT-4 and GPT-4o spot the conflict while the other two models still lag. On the other hand, results are reasonable for two categories of name vs. gender and parenthood vs. marital status. For name vs. gender, GPT-4o is superior with only one incorrect prediction. One error in this category, correctly identified only by GPT-4o, involves the utterance: 'I am not a girl. My name is Faranak.' 'Faranak' is a typically female name and, without diacritics, can be mistakenly interpreted as 'Frank', a typical Western male name. To determine if the error relates to this specific name or the structure containing a denial of one gender and assignment of another, we modified it to 'I am not a girl. My name is Mohammad.' This version contains no conflict, yet all models except GPT-4o erroneously reasoned that, since it denies being a girl, assigning a male name creates a contradiction.

For parenthood vs. marital status, the results are promising with just one error in GPT-3.5 and GPT-4 while Llama and GPT-4o detect all correctly. For Llama, it relates to its bias toward inferring marriage from having children. The erroneous case here is: 'Although I have never been a groom, I have 3 beautiful daughters.' GPT-4 correctly associated 'groom' with a wedding and interpret it as getting married without a ceremony while GPT-3.5 wrongly associate it to son-in-law and found a false mismatch.

In general, Llama and GPT-3.5 are more prone to false conflict detection by interpreting the granularity of residence as a conflict: When a user specifies a broad geographical region and then narrows it to a specific city or province, the model detects it as a contradiction. They even interpret questions about the other user's residence, such as 'where are you from' at the end of the utterance, implying a difference between the speaker's stated precise location and the listener's unspecified residence.

Llama also shows biases toward specific names. For instance, in the utterance 'My name is Aryan. I am a self-made woman...', the model associates the male name 'Aryan' /aːriˈjaːn/ with traditional cultural roots, contrasting it with more modern values of a 'self-made woman'. Similarly, in 'I am Simanah. I am 20 years old now, and I am a nurse at Mehr

Hospital', it perceives 'Simanah' as an old-fashioned name, suggesting a more mature person than 20. In 'My name is Pegah, and I became a mother last year.', it interprets 'Pegah' as a name for a young person, conflicting with the role of a mother as more responsible and mature person. Additionally, Llama-70b shows biases in assigning specific jobs to men (e.g., optometrist) or viewing mechanical engineering as a profession requiring a strong financial background. These biases indicate the need for caution in using this model to avoid misinformation.

## 6.2 Analyzing zero-shot results of LLMs for slot filling

Our analysis of LLM outputs reported in Table 9, uncovered that although Llama was instructed to find slot values in user's utterance, many errors stem from extracting values from the previous chatbot utterance. Also, both models require clearer definitions for the "hobby" slot type. They often misinterpret expressions such as liking or being interested in something as a hobby (e.g., 'I like to pursue my goals in creating employment opportunities and helping people find better jobs') or mistakenly identify the user's favorite food as a hobby (e.g., 'I am very interested in Chinese food; it is delicious.'). In some instances, they even treat questions about the other user's hobby as their own hobby (e.g., 'I have read that book too. What movies do you watch?', hobby = reading books & watching movies). There are also cases of entirely incorrect assignments (e.g., 'I have always strived to protect the environment. It is very important to preserve this ecosystem for the next generation.', hobby = environment protection or 'I want to work on the development of software for running.', hobby = running).

For the "residence" slot type, models often mistakenly identify the city or province mentioned in the conversation as the user's residence, even if it's just mentioned in question or related to weather topic (e.g., 'Hi, are you from Tehran?' or 'I also love the summer in Rasht. The weather is very mild.').

Regarding the "job" slot, there are instances where GPT-4o incorrectly assumes the user's education (e.g., 'I studied electrical engineering. What education did you pursue after high school?' inferred as 'Electrical Engineer') or GPT-4o and Llama both infer general expressions as user's job title (e.g., 'No, I am working. What are you doing?' inferred as 'employee' in GPT-4o and 'working' in Llama while 'Yes, I am working too. Are you a homemaker?' inferred as 'manual worker' in GPT-4o and as 'they work' in Llama). GPT-4o occasionally assumes gender based on specific job titles (e.g., 'internal designer' or 'sculptor' inferred as the user being 'male'), whereas there is a consistent pattern of bias in Llama by for instance associating certain professions. Additionally, Llama frequently infers marital status from having children, a trait rarely seen in GPT-4o and both models correctly identify gender based on well-known gender-associated names.

Both GPT-4o and Llama lag in calculating age based on birth date. In GPT-4o, this is due to incorrect calculations, while in Llama, it results from not knowing the current year of the solar calendar.

For intent detection, both models do not adhere to our predefined topics. Cases with the topic "other" are assigned new, descriptive titles like "effort to lose weight" or "respecting others' interests". This noncompliance and the strict requirement to include all topics in multi-intent cases have decreased intent accuracy in both models.

## 6.3 Analysis of zero-shot results with refined prompt

The analysis of the results related to the Phi-4 model, in zero-shot learning with our refined prompt, suggests that it struggles with Persian verb conjugation. In some cases, it outputs the exact inflected phrase as the slot value (such as "فیلم میبینم" /fiːlm mibinæm/ ("I watch movies") for "hobby") and even some times, it incorrectly constructs the infinitive form of the verb in the phrase (such as "بلدن زبان جاوا" /bælædæn-e zæbaːn-e dʒaːvaː/ (known java language) instead of "بلد بودن زبان جاوا" /bælæd budæn-e zæbaːn-e dʒaːvaː/ (knowing java language), "نوشیدن غذاهای محلی" /nuːʃidæn-e Gæzaːhaːj-e mæhælliː/ (drinking local food) instead of "نوش جان کردن غذاهای محلی" /nuːʃe dʒaːn kærdæn-e Gæzaːhaːj-e mæhælliː/ ("eating local foods"), or "گردشگری" /gærdeʃgæriː/ (tourism) instead of "گردش کردن" /gærdeʃ kærdæn/ ("strolling") for "hobby" slot).

Additionally, compared to GPT-4o and Llama-70B, the model is worse in following instructions correctly. Despite being explicitly instructed not to interpret the user's workplace as their residence, it still treats "Sa'dabad Palace" or "protected areas" as a residence in the utterances "I am the museum guide of Sa'dabad Palace" and "I am a park ranger in protected areas.." It should be noted that, like other LLMs, this model also struggles with recognizing the number of children and siblings. Specifically, it fails to interpret phrases indicating the absence of children or siblings (such as generating "I don't know" as the number of siblings instead of "0 sisters" for the utterance "I have no sister.")

Regarding the "name" slot, the model is significantly weak in recognizing Persian names. Unlike GPT-4o, which can even infer gender from names, Phi-4 struggles to detect them, especially when presented in the form of "I am #Name" rather than the more explicit form of "My name is #Name." In such cases, it often misclassifies the name to the another slot type, particularly "job." For instance, in the utterance "I am Karen," the model incorrectly assigns "Karen" (/kaːrɛn/), a male name, to the "job" slot due to its orthographic similarity to "kar" (/kaːr/), which means job in Persian. Similarly, it assigns "Shahrouz" (/ʃæhˈruːz/) to the "residence" because of its similarity to "shahr" (/ʃæhr/), meaning city, in the utterance "I am Shahrouz. Where are you from?." Even more surprisingly, it assigns "Pegah" /peˈgaːh/ (پگاه) to the "job," possibly due to mistakenly translating its orthographic form and associating it with "guard" in Persian (نگاهبان /neˈgaːhbaːn/).

Also, Phi-4 underperforms in recognizing Iranian cities and provinces compared to GPT-4o and Llama-70B, especially when affixes acting as demonyms accompany these locations. For example, Phi-4 generates "شوشه" /ʃuːʃe/ instead of "شوش" /ʃuːʃ/ in the utterance "من اهل شوشم" /mæn ʔæhl-e ʃuːʃ-æm/ ("I am from Shush"), or fails to recognize "Rasht" in the utterance "من رشتی هستم" /mæn ræʃti hæstæm/ ("I am from Rasht").

Regarding the 'job' slot, model behaves like the others and assigns general or broad values to the slot such as "IT company," "business," or the very generic term of "work" in the following utterances: "I work in an IT company", "I am in the field of business" and "I am working".

For 'gender' assignment, model exhibits two types of errors. First, it makes completely irrelevant biases towards certain expressions, such as "I am #age years old" in utterances like "I'm 27 years old. Where are you from?" or "Yes, sure. I'm 26 years old and I'm from a town in Mazandaran." According to the model's explanation, this lead to incorrect inference of 'female' value for the 'gender' slot.

There are also cases where the model completely make mistakes and assigns semantically incorrect values to the slots such as assigning "education" value to 'hobby' slot in this utterance: "I am studying computer science. What is your occupation?" or assigning "job" to 'job' slot in the utterance of "My plans for the future are very broad. the first step is finding a good job."

On the other hand, the analysis of Llama-3.1-8B results indicates a strong bias in irrelevant assigning of value of "single" or in some cases, "married" to the slot of 'marital status'. When asked for an explanation in some instances, the model either provides an irrelevant justification, which, at best, could be considered biased or refers to a text that does not exist in the conversation and indicate hallucinatory reasoning. For example, it attributes "single" to this slot in following dialogue: "-Yes, that's right. What's new with you? -Everything is good. But my job has become a bit harder." with this reasoning: "In this response, because the user said 'everything is good,' and this sentence generally refers to the user's overall good situation in life, we have categorized the user as 'single'".

The model demonstrates a clear tendency to assign hobby values to both the 'hobby' and 'job' slots (such as categorizing "watching TV series" as both a hobby and a profession in the utterance "I love watching TV series"). Moreover, the model frequently interprets question phrases as values for different slot types (e.g., assigning "Where do you live?" to 'residence' or "Where do I know you from?" to the 'name' slot). In some cases, it even misclassifies pronouns like "I" as a name or "my hometown" as a specific location for 'residence'.

Also, a further bias of the model is regarding "city" as "Tehran" in several instances, such as assuming the value of "Tehran" for 'residence' slot in the utterances like "I'm Mehdi. Which city do you live in?" or "Everything is fine, thanks. Which city do you live in?". The last source of error in this model, relates to assigning completely irrelevant or illogical values,

such as "friendship" to the 'hobby' slot in the utterance: "I hope a good <u>friendship</u> forms between us" or assigning "sister, brother" to the 'hobby' in the utterance: "Sometimes having a <u>sister</u> or <u>brother</u> is the greatest wealth.".

In general, the analysis of the results from all LLMs employed in our research reveals that they are all unable to extract the specified values for the number of children or siblings, particularly when their absence is indicated with phrases like "I have no siblings/children." When two or three values for the number of sisters, brothers, or total siblings (or children) are mentioned in an utterance, models can only extract one or at most two of these values, failing to extract all of them. Also, another type of error relates to when the user provides an acceptable value for a slot in the form of a question (e.g.,"Are you from <u>Shahrud</u>?" (residence: "Shahrud")). The most frequent general error, however, is extracting values from the previous chatbot utterance in the conversation instead of extracting from the user's current utterance; Despite the explicit instruction in the prompt that the previous utterance refers to the chatbot, except GPT-4o, which occasionally makes this mistake, all the other models consistently fail to follow it.

## 6.4 Analyzing the results of fine-tuned LLM for slot filling

Regarding all of these shortages, we fine-tune Phi-4 which shows substantial improvements compared to zero-shot method with 18.2% increase in slot-F1. To find a more fine-grained understanding of the model's performance, we examine improvements across different slot types with F1 score shown in Fig 7. we also include the zero-shot results of other models for comparison.

Llama-3.1-8B has the lowest score across all slot categories. For the 'hobby' slot, LLMs generally perform better, suggesting that this category is the most easily recognized by them. The fine-tuned model outperforms GPT-4o which appears to be more restricted in classifying interests as hobbies. For the 'job' slot, our trained model achieves the best performance while GPT-4o performs better than the fine-tuned model. This is due to the fine-tuned model's tendency to incorrectly classify general phrases (e.g., 'I work' or 'working') as job titles, one of Phi-4's inherent issues that is not fully resolved through fine-tuning. Also, the fine-tuned model often misclassifies educational values as job titles (e.g., 'computer science' in the utterance 'I study <u>computer science</u>.'). For the 'gender' slot, the fine-tuned model performs best, as it has learned to avoid making inferences based on other slot values while GPT-4o is more relied on inferred information and may introduce biases. For 'marital status', Llama-3.1-8B with strong inherent biases, has the weakest performance while the fine-tuned model performs best. The difference between GPT-4o and the fine-tuned model stems from a single case where GPT-4o mistakenly extracts value from the previous chatbot utterance. The discrepancy between our module and other LLMs arises from two cases: one where the model incorrectly extracts a value from a question ('Are you <u>married</u>?')

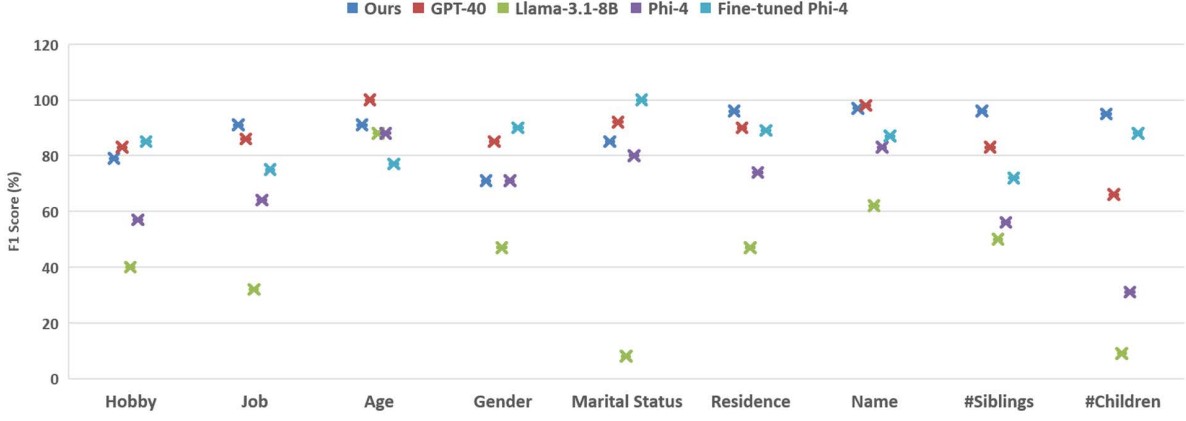

**Fig 7. Comparing the Performance of our slot filler Across Different Slot Types using F1 score.**

and another where it misinterprets the phrase 'I am <u>single</u>, but I am going to get <u>married</u> soon', extracting both 'married' and 'single'. For 'residence', our module performs best, while GPT-4o and the fine-tuned model have similar but slightly lower performance. Their lower scores are due to a tendency to extract city names from unrelated contexts, such as weather discussions or to incorrectly retrieve residence information from previous system utterances. For the 'name' slot, as previously mentioned, Phi-4 struggles with recognizing Persian names while fine-tuning improves this but its performance remains below GPT-4o. Our method performs similarly to GPT-4o which ranks highest in this category. Finally, for the 'number of siblings' and 'number of children', LLMs generally struggle to accurately detect all mentioned numbers. Our method outperforms others, achieving a significant 13% increase in F1 for siblings and 7% increase for the number of children compared to the next-best LLM model.

## 7  Literature review

### 7.1  Personal profile extraction

Recently, there has been a great deal of interest in finding personal information in different applications. Early methods [56,57] use traditional machine learning techniques to tackle this problem. Some approaches cast the task as predicate-argument structure identification [9]. With the advent of deep learning, more sophisticated methods have emerged, including the use of variational autoencoders [10], techniques for learning latent speaker representations [58,59], and the application of Language Models [60–62]. Clearly, work has particularly focused on leveraging pretrained language models [63], with the latest approaches exploring the potential of large language models (LLMs) [64,65]. For example, Villa Monedero et al. [66] frame the task as triplet extraction and utilize models such as Phi series [67], Llama3-8B [68], and Gemma2 [69] in both zero-shot and fine-tuned settings, achieving promising results in English. Despite this progress, there is still significant room to explore the capabilities of LLMs in low-resource languages.

From the perspective of data extraction domains, a group of studies [70,71] have focused on social media content, which differs from natural, short conversational texts in dialogue systems and contains additional signals such as user activity or hashtags. Moreover, they typically extract attributes with binary or small sets of values. Tigunova [7] proposes two deep learning-based architectures: Hidden Attribute Models (HAMs) and Retrieval-based Hidden Attribute Models (RHAMs), focusing only on extracting two nearly open-set attributes—job and hobby—while still relying on social content from Reddit submissions.

As mentioned earlier in §1, although some progress has been made in developing Persian NLU datasets and models [24–29], there is limited research in the specific area of user profile extraction. The work of Safari and Shamsfard [43] is, to our knowledge, the only study focused solely on data curation for user personal profile. Our contribution lies in combining classical approaches with cutting-edge LLMs, setting our work apart from prior efforts.

### 7.2  Slot filling & intent detection methods

Regarding the extractable information as slots and the topic of the utterance as intent, we also review the main approaches in Intent Detection (ID) and Slot Filling (SF) methods. Early research examined ID and SF separately or sequentially using a pipeline model. The main drawback of these approaches is their susceptibility to error propagation and the need for two separate models for each subtask [72].

Subsequent studies noted that using auxiliary tasks in multi-task learning [73–75], specifically training SF and ID jointly, can be more beneficial due to their mutual influence. Most joint training methods, regarding the sequential nature of SF, employ recursive structures (RNNs, GRUs, or BiLSTMs) with attention mechanism and various word and character embeddings. Liu and Lane [76] used a BiLSTM as a shared encoder and two task-specific LSTMs to jointly train ID and SF. Qin et al. [47] proposed a hierarchical format to more explicitly use intent information for slot filling. Using a self-attentive BiLSTM module as the shared encoder, they performed ID is at the token level and concatenated the ID label

as an input feature for the SF task. Wang et al. [77] proposed a Bi-model trained asynchronously using two separate cost functions and two separate BiLSTMs that accept the hidden states of the other task as input features.

With the emergence of transformers, some researchers have replaced the recursive structure with this architecture to more effectively address long-distance dependencies. Qin et al. [78] employed a shared BiLSTM as an encoder and replaced core RNN modules with a transformer-based architecture. It consists of a label attention for each task followed by a co-interactive attention mechanism to capture their interaction. At the end, output vectors are passed to separate decoders which is a max-pooling layer for ID and a CRF for the SF task. Chen et al. [31] employed BERT language model [46], a bidirectional transformer encoder with attention mechanisms, to obtain the final hidden states of input tokens and pass them through two task-specific layers with the objective function of maximizing the joint conditional probability of both tasks.

The advent of Large Language Models (LLMs) intrigue some researches to investigate their ability through zero or few-shot learning method [48,49,79]. As an example, in the study of Zhu et al. [50], they have proposed a zero-shot technique in a two steps. First, they have asked ChatGPT to find the intent and slots of the input sentence and then, mutually fed predictions into prompts of the other task to leverage their correlation.

However, LLMs often struggle with maintaining factual consistency and up-to-date information [80–82] and they are also prone to generating hallucination in a confident tone [83]. As a result, a growing number of researches [84–87] integrate domain-specific ontologies into chatbot reasoning which is a more reliable approach. It highlights the value of combining structured, context-aware knowledge to improve system's consistency and accuracy while reducing errors like hallucinations.

## 8 Conclusion

In this research, we have presented a multi-step approach to extract both explicitly and implicitly expressed personal information, such as name or occupation, from user dialogues. It can even detect semantic or pragmatic contrasts through conversation. We have introduced a number of techniques to improve our multi-step approach and surpass the performance of large language models. In our future work, we will inject adversarial examples to the training set to improve the robustness of our slot filling module in information extraction. Moreover, we will incorporate the context of the previous utterance in the first step to make the model more context-aware in detecting personal attributes.

## Supporting information

**S1 File. Supplemental Figures and Tables.** All supporting figures and tables referenced throughout the text are located in this file.
(PDF)

## Author contributions

**Conceptualization:** Pegah Safari, Mehrnoush Shamsfard.

**Formal analysis:** Pegah Safari.

**Investigation:** Pegah Safari.

**Methodology:** Pegah Safari, Mehrnoush Shamsfard.

**Project administration:** Mehrnoush Shamsfard.

**Software:** Pegah Safari.

**Supervision:** Mehrnoush Shamsfard.

**Validation:** Pegah Safari, Mehrnoush Shamsfard.



**Writing – original draft:** Pegah Safari.

**Writing – review & editing:** Mehrnoush Shamsfard.

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
