## [Decision Letter · Decision Letter 0]

26 Jun 2025

PONE-D-25-19130Extracting User Profile via Large Language Models and OntologiesPLOS ONE

Dear Dr. Shamsfard,

Thank you for submitting your manuscript to PLOS ONE. After careful consideration, we feel that it has merit but does not fully meet PLOS ONE’s publication criteria as it currently stands. Therefore, we invite you to submit a revised version of the manuscript that addresses the points raised during the review process.

**Please add in required details mentioned in reviewer comments.**

We look forward to receiving your revised manuscript.

Kind regards,

Ying Shen, Ph.D.

Academic Editor

PLOS ONE

**Journal Requirements:**

1. When submitting your revision, we need you to address these additional requirements. Please ensure that your manuscript meets PLOS ONE's style requirements, including those for file naming. The PLOS ONE style templates can be found at https://journals.plos.org/plosone/s/file?id=wjVg/PLOSOne_formatting_sample_main_body.pdf and https://journals.plos.org/plosone/s/file?id=ba62/PLOSOne_formatting_sample_title_authors_affiliations.pdf 2. Thank you for uploading your study's underlying data set. Unfortunately, the repository you have noted in your Data Availability statement does not qualify as an acceptable data repository according to PLOS's standards. At this time, please upload the minimal data set necessary to replicate your study's findings to a stable, public repository (such as figshare or Dryad) and provide us with the relevant URLs, DOIs, or accession numbers that may be used to access these data. For a list of recommended repositories and additional information on PLOS standards for data deposition, please see https://journals.plos.org/plosone/s/recommended-repositories.

Reviewers' comments:

Reviewer's Responses to Questions

**Comments to the Author**

1. Is the manuscript technically sound, and do the data support the conclusions?

Reviewer #1: Yes

Reviewer #2: Yes

2. Has the statistical analysis been performed appropriately and rigorously? 

Reviewer #1: I Don't Know

Reviewer #2: Yes

3. Have the authors made all data underlying the findings in their manuscript fully available?

Reviewer #1: Yes

Reviewer #2: Yes

4. Is the manuscript presented in an intelligible fashion and written in standard English?

Reviewer #1: Yes

Reviewer #2: Yes

5. Review Comments to the Author

**Reviewer #1:**  The journal submission "Extracting User Profile via Large Language Models and Ontologies" presents an information extraction work where the authors infer personal information of Persian speakers from their conversations with chatbots. The authors propose a multi-step approach involving prompting, slot-filling and ontology-mapping to find the user profile. Experiments show that the proposed approach performs much better than baseline zero-shot and few-shot prompting methods.

The authors should clarify the distinction between single-intent and multi-intent modes. An utterance could convey multiple intents and each intent belongs to a finite set of intents. Therefore, intent detection becomes a multi-label classification task and the classifier should model k separate probabilities, where k = 13. However, the model architecture uses a single sigmoid or softmax layer. I recommend the authors should clarify their model description.

While the authors do not need to perform this experiment, I would much appreciate their thoughts on why they did not fully write their prompts in Persian. Currently, only the demonstrations are in Persian but the instructions are in English.

I would recommend the authors change the subsection headers of the discussion section by removing the word "ablation". The authors are qualitatively evaluating the results of their models' outputs. This is different from ablation studies which includes evaluating the performance when we suppress some subcomponent in our model. The authors could also try minimizing their discussion sections and only providing the principal insights; the discussion content is too detailed and specific, and probably would not generalize to other datasets.

In lines 143-145, the authors say they use separate evaluation and test sets. I think what the authors are referring to as the evaluation set is actually the development or validation set which is used to pick the best model parameters. I recommend using the more conventional name: development or validation set.

The authors should also mention the statistical tests they performed to compare performances of different methods.

This work makes an important contribution towards modeling in low-resource languages. I believe some minor revisions, as recommended above, could greatly strengthen the manuscript.

**Reviewer #2:**  This paper proposes a multi-step method for extracting user profiles from Polish dialogues. Specifically, the approach employs a multi-task learning model combining slot filling and intent detection to extract phrases that reveal user attributes. In parallel, a large language model (LLM) is used in a few-shot in-context learning setup to infer more complex user attributes from the same utterances. Finally, the outputs of both the slot filler and the LLM are integrated into a designed ontology to construct the final user profiles. This approach enables the transformation of unstructured information in free-text into structured concepts, and allows for the detection of potential semantic inconsistencies. The authors further validate the superiority of their method through comparative experiments against strong baselines such as GPT-4o and LLaMA-3-70B.

Overall, the paper is well-written and the research is solid. However, I have the following comments and suggestions for improvement:

1. **Lines 73–79**: The use of variables *x*, *y*, and *h* is confusing, as *xᵢ*, *yᵢ*, and *hᵢ* already represent tokens, their labels, and their vector representations, respectively. Please consider using different notations to avoid ambiguity.

2. **Section 3.2 – Slot Filling**: I recommend including a figure that illustrates the slot filling process described in this section, along with concrete input-output examples to help readers better understand the model workflow.

3. **Line 95**: Please clarify how the positive and negative samples were constructed. Specifically, for how many target concepts were these samples created?

4. **Line 96**: Which LLM is used in this process?

5. **Line 100 – Step 2: Ontology-based Information Extraction**: The construction of the ontology should be described here rather than being deferred to Section 3.4. Additionally, it would be helpful to include a table or diagram illustrating the hierarchical structure of concepts in the ontology.

6. **Line 164**: The authors mention that "in our prompt, we directly ask about each of our nine desired personal information." Please enumerate what these nine items are.

7. **Section 4 – Experimental Results**: I suggest including a case study to illustrate the outputs generated by different models. Accompanying content analysis would be beneficial to highlight the strengths and limitations of each model.

6. PLOS authors have the option to publish the peer review history of their article (what does this mean? ). If published, this will include your full peer review and any attached files.

**Do you want your identity to be public for this peer review?** For information about this choice, including consent withdrawal, please see our Privacy Policy .

Reviewer #1: **Yes:** Sabyasachee Baruah

Reviewer #2: No

---

## [Author Response · Author response to Decision Letter 1]

10 Jul 2025

We sincerely thanks the editor and the reviewers for the time and effort spent for reviewing our manuscript We appreciate the constructive comments and suggestions which helped us improve the clarity and the quality of our manuscript.

the point-by-point response to each of the reviewers' comments is as follows:

Reviewer #1:

1. The authors should clarify the distinction between single-intent and multi-intent modes. An utterance could convey multiple intents and each intent belongs to a finite set of intents. Therefore, intent detection becomes a multi-label classification task and the classifier should model k separate probabilities, where k = 13. However, the model architecture uses a single sigmoid or softmax layer. I recommend the authors should clarify their model description.

Answer: Thanks for your observation. After pooling hidden vectors, we apply a fully connected layer as our intent classifier with k=13 outputs. On top of these outputs which represent intent logits, we then apply two separate activations: softmax for single-intent and sigmoid for multi-intent classification.

So, we added more clarifying descriptions to the end of the ‘Multitask Learning with Intent Detection’ section (in lines 85–93).

2. While the authors do not need to perform this experiment, I would much appreciate their thoughts on why they did not fully write their prompts in Persian. Currently, only the demonstrations are in Persian but the instructions are in English.

Answer: Thank you for your question. We intentionally used English for prompting the LLMs while keeping the context and input utterances in Persian. This approach minimizes the influence of the prompt language on model performance, ensuring that only the input language (Persian) affects the outcome. Prior studies [1, 2] have also shown that prompting multilingual LLMs in English is generally more effective than using languages like Persian. Additionally, translating well-established terms such as "intent" into Persian can introduce ambiguity or translation errors for the model. Using English prompts also improves the comparability of our work, allowing future researchers to reuse the same prompt format by substituting only the input utterances in their target language.

We added this description to the article in section 3.3 (in lines 228-232).

3. I would recommend the authors change the subsection headers of the discussion section by removing the word "ablation". The authors are qualitatively evaluating the results of their models' outputs. This is different from ablation studies which includes evaluating the performance when we suppress some subcomponent in our model. The authors could also try minimizing their discussion sections and only providing the principal insights; the discussion content is too detailed and specific, and probably would not generalize to other datasets.

Answer: As recommended, we have replaced the term ‘ablation’ (which is more related to section 5) with ‘analysis’ in the subsection titles. Additionally, we revised the discussion sections to present more concise and focused content by highlighting the key points and removing any overly specific details.

4. In lines 143-145, the authors say they use separate evaluation and test sets. I think what the authors are referring to as the evaluation set is actually the development or validation set which is used to pick the best model parameters. I recommend using the more conventional name: development or validation set.

Answer: Thanks. Since it may be misinterpreted, we replaced two occurrences of ‘evaluation’ with ‘validation’ in section 3.1.

5. The authors should also mention the statistical tests they performed to compare performances of different methods.

Answer: Thank you for your comment. We performed paired t-tests over the five folds to assess the statistical significance of our method against LLMs which results confirm the superiority of our method over both Llama-3-70B (t(4) = 39.10, p < 0.001) and GPT-4o (t(4) = 29.50, p < 0.001). We added these statistical test results in the section 4.1 (in lines 242-244).

Reviewer #2:

1. **Lines 73–79**: The use of variables *x*, *y*, and *h* is confusing, as *xᵢ*, *yᵢ*, and *hᵢ* already represent tokens, their labels, and their vector representations, respectively. Please consider using different notations to avoid ambiguity.

Answer: Thanks for your comment. To avoid ambiguity, we use ‘X’ to represent the input sentence, and ‘Y’ and ‘H’ to clearly distinguish between the label vector components (‘yᵢ’) and the hidden state representations (‘hᵢ’), respectively.

2. **Section 3.2 – Slot Filling**: I recommend including a figure that illustrates the slot filling process described in this section, along with concrete input-output examples to help readers better understand the model workflow.

Answer: Thanks for your suggestion. To illustrate the overall flow of input utterance and the system output, we include Figure 5 at the end of section 2 (‘our approach’) that demonstrate suitable example.

3. **Line 95**: Please clarify how the positive and negative samples were constructed. Specifically, for how many target concepts were these samples created?

Answer: These examples are heuristically constructed with a complex sentence structure and inclusion of idioms. We build these examples for two main concepts in the ontology alongside their 25 sub-concepts related to family roles and educational statuses. In order to make this process clearer, we add more description in lines 101-106.

4. **Line 96**: Which LLM is used in this process?

Answer: The LLM model utilized in the process is ‘Llama-3-70B’. To add more clarification, we explicitly mentioned the model’s name in line 109.

5. **Line 100 – Step 2: Ontology-based Information Extraction**: The construction of the ontology should be described here rather than being deferred to Section 3.4. Additionally, it would be helpful to include a table or diagram illustrating the hierarchical structure of concepts in the ontology.

Answer: For better alignment with the presentation of our method, as you recommended, we have moved the description of the ontology structure from Section 3.4 to the section of ‘Step 2: Ontology-based Information Extraction’. Also, following your helpful suggestion, we have included Figure 4 to illustrate a simplified view of the concepts along with a subset of their relationships. It is noteworthy that the ontology is more complicated than its depiction in Figure 4.

6. **Line 164**: The authors mention that "in our prompt, we directly ask about each of our nine desired personal information." Please enumerate what these nine items are.

Answer: These nine items include name, age, gender, marital status, occupation, hobby, residence, number of children, and number of siblings. To improve clarity, we explicitly listed these nine categories of personal information in this section (in lines 220-221).

7. **Section 4 – Experimental Results**: I suggest including a case study to illustrate the outputs generated by different models. Accompanying content analysis would be beneficial to highlight the strengths and limitations of each model.

Answer: Thanks for your comment. In Section 6 (‘Discussion’), particularly in the first two subsections of 6.1, we already have provided a detailed analysis of the results from our proposed approach and the LLM outputs for profile extraction and conflict detection. Also, we have examined the outputs thoroughly and have highlighted the strengths and limitations of the models. However, as you suggested and to further enhance understanding of the models’ outputs, we included Table 13 as the case study at the end of section 6.1.

Comment references:

1. Jin Y, Chandra M, Verma G, Hu Y, De Choudhury M, Kumar S. Better to ask in English: Cross-lingual evaluation of large language models for healthcare queries. In: Proceedings of the ACM Web Conference 2024; 2024. p. 2627–2638.

2. Myung J, Lee N, Zhou Y, Jin J, Putri R, Antypas D, et al. Blend: A benchmark for LLMs on everyday knowledge in diverse cultures and languages. Advances in Neural Information Processing Systems. 2024; 37:78104–78146.

Also, in response to the journal's requirements mentioned in the letter, we reviewed our paper to ensure compliance with the journal template and submitted our data to the Zenodo repository (with DOI: 10.5281/zenodo.15839326).

---

## [Decision Letter · Decision Letter 1]

24 Jul 2025

Extracting User Profile via Large Language Models and Ontologies

PONE-D-25-19130R1

Dear Mehrnoush Shamsfard,

We’re pleased to inform you that your manuscript has been judged scientifically suitable for publication and will be formally accepted for publication once it meets all outstanding technical requirements.

Kind regards,

Ying Shen, Ph.D.

Academic Editor

PLOS ONE

Additional Editor Comments (optional):

Reviewers' comments:

Reviewer's Responses to Questions

**Comments to the Author**

1. If the authors have adequately addressed your comments raised in a previous round of review and you feel that this manuscript is now acceptable for publication, you may indicate that here to bypass the “Comments to the Author” section, enter your conflict of interest statement in the “Confidential to Editor” section, and submit your "Accept" recommendation.

Reviewer #2: (No Response)

2. Is the manuscript technically sound, and do the data support the conclusions?

Reviewer #2: (No Response)

3. Has the statistical analysis been performed appropriately and rigorously? 

Reviewer #2: (No Response)

4. Have the authors made all data underlying the findings in their manuscript fully available?

Reviewer #2: (No Response)

5. Is the manuscript presented in an intelligible fashion and written in standard English?

Reviewer #2: (No Response)

6. Review Comments to the Author

Reviewer #2: (No Response)

7. PLOS authors have the option to publish the peer review history of their article (what does this mean? ). If published, this will include your full peer review and any attached files.

**Do you want your identity to be public for this peer review?** For information about this choice, including consent withdrawal, please see our Privacy Policy .

Reviewer #2: No

---

## [Editor Report · Acceptance letter]

PONE-D-25-19130R1

PLOS ONE

Dear Dr. Shamsfard,

I'm pleased to inform you that your manuscript has been deemed suitable for publication in PLOS ONE. Congratulations! Your manuscript is now being handed over to our production team.

Kind regards,

on behalf of

Dr. Ying Shen

Academic Editor

PLOS ONE